# Safety of changes in the use of noninvasive ventilation and high flow oxygen therapy on reintubation in a surgical intensive care unit: A retrospective cohort study

Stanislas Abrard[1,2,3]⊛*, Lorine Jean[1]⊛, Emmanuel Rineau[1,2], Pauline Dupré[1], Maxime Léger[1,4], Sigismond Lasocki[1,2]

1 Department of Anesthesiology and Intensive Care, University Hospital of Angers, Angers, France,
2 MITOVASC Institut, INSERM 1083—CNRS 6015, University of Angers, Angers, France, 3 Department of Anesthesiology and Critical Care Medicine, Edouard Herriot hospital, Hospices Civils de Lyon, Lyon, France,
4 INSERM UMR 1246—SPHERE, Nantes University, Tours University, Nantes, France

⊛ These authors contributed equally to this work.
* stanislas.abrard@chu-lyon.fr

**Data Availability Statement:** All relevant data are within the paper and its Supporting Information files.

## Abstract

Reintubation after weaning from mechanical ventilation is relatively common and is associated with poor outcomes. Different methods to decrease the reintubation rate post extubation, including noninvasive ventilation, and more recently high-flow oxygen (HFO) therapy, have been proposed. In this study, we aimed to assess the safety of introducing HFO in the post-extubation care of intensive care unit (ICU) patients. We conducted a single-center cohort study of extubated adult patients hospitalized in a surgical ICU and previously mechanically ventilated for > 1 day. Our study consisted of two phases: Phase 1 (before the introduction of HFO from April 2015 to April 2016) and Phase P2 (after the introduction of HFO from April 2017 to April 2018). The primary endpoint was the reintubation rate within 48 hours of extubation. In total, 290 patients (median age 65 years [50–74]; 190 men [65.5%]) were included in the analysis (181 and 109 in Phases 1 and 2, respectively). The results of the post-extubation use of noninvasive methods (noninvasive ventilation and/or HFO) were not significantly different between the two phases (41 [22.7%] versus 29 [26.6%] patients; p = 0.480), however these methods were implemented earlier in Phase 2 (0 versus 4 hours; p = 0.009) and HFO was used significantly more often than noninvasive ventilation (24 [22.0%] versus 25 [13.8%] patients; p = 0.039). The need for reintubation within 48 hours post extubation was significantly lower in Phase 2 (4 [3.7%] versus 20 [11.0%] patients; p = 0.028) but was not significantly different at 7 days post extubation (10 [9.2%] versus 30 [16.6%] patients; p = 0.082). The earlier implementation of noninvasive methods and the increased use of HFO beginning in Phase 2 were safe and effective based on the reintubation rates within the first 48 hours post extubation and after 7 days.

**Funding:** The author(s) received no specific funding for this work.

**Competing interests:** The authors have declared that no competing interests exist.

## Introduction

In intensive care units (ICUs), patients weaned from invasive mechanical ventilation are at risk of post-extubation respiratory failure (acute respiratory failure [ARF]) and subsequent mechanical ventilation with tracheal reintubation [1,2]. Approximately 10–15% of patients are reintubated within the first 48 hours following extubation [3], with an associated increase in morbidity, mortality, and length of hospital stay [4]. The causes of respiratory failure after extubation include upper airway obstruction, inadequate cough, atelectasis, encephalopathy, and cardiac dysfunction [2,5–7]. Extubation failure is currently defined as the need for reintubation within 48 hours of extubation [8,9]. Three noninvasive methods are available to prevent and/or treat post-extubation ARF: conventional oxygen therapy, high-flow conditioned oxygen therapy (HFO), and noninvasive ventilation (NIV). NIV is one of the recommended treatments for acute hypercapnic and hypoxemic ARF [10]. NIV administered in the post-extubation period can recruit zones of alveolar collapse [11,12], improve oxygenation [13], and minimize the work of breathing [1,14]. Although the use of NIV as supportive treatment to avoid tracheal reintubation is ineffective in 10–50% of patients with post-extubation ARF, two meta-analyses concluded that the early use of NIV could decrease reintubation rates [15,16]. HFO, a newly developed technology that delivers a high flow of high-concentration oxygen via nasal cannula, is able to generate mild continuous positive airway pressure [17], which reduces the work of breathing and clears out upper airway dead space [18]. Compared with conventional oxygen therapy, HFO therapy after extubation improves oxygenation and patient comfort and prevents post-extubation ARF and reintubation in general populations of critically ill patients [3,19]. Unlike NIV, HFO does not provide the continuous positive airway pressure that supports the patient's inspiratory efforts. Two prospective randomized controlled studies compared post-extubation NIV with HFO and found no difference in reintubation rates in critically ill medical and surgical patients [20,21]. Analyses of these data, pooled in a meta-analysis, showed no significant difference in clinical outcomes between NIV and HFO [22]. However, there is currently limited data on the supportive use of a combination of HFO and NIV for the treatment of post-extubation ARF. Moreover, the use of noninvasive methods has raised safety concerns [23]. These therapies might increase the risk of poor outcomes due to the apparent improvements of patient comfort and oxygenation leading to delayed reintubation.

In our surgical ICU, the potential benefit of HFO has led to a change in post-extubation practices, with more frequent use of HFO in the post-extubation period. The main objective of this study was to assess the safety of this practice change using the rate of reintubation 48 hours after extubation (compared between the two periods, before and after the introduction of HFO) as the primary endpoint.

## Methods

### Study design and setting

This retrospective cohort study, ARVENiO (Analyse Rétrospective de l'utilisation de la Ventilation Non-invasive et de l'Oxygénothérapie), was approved by the institutional review board of the French Anesthesiology and Critical Care Society (SFAR; 74 rue Raynouard; 75016 Paris France; Chairperson Pr JE. Bazin) on May 27, 2019 (Ethical Committee N°IRB 00010254-2019-094) and was registered at the National Commission for Information Technology and Civil Liberties (DRCI-CGDE-FO-005), according to French law [24]. During their hospital stay, patients were advised that their data could be used retrospectively for clinical research purposes. Written informed consent is generally waived for observational retrospective

studies; however, patients had the right to refuse the use of their data upon request (none of them did) [24]. Our study is reported according to the Strengthening the Reporting of Observational studies in Epidemiology guidelines for reporting observational studies and was conducted in accordance with the Declaration of Helsinki.

This single-center, historical cohort was conducted in our 12-bed surgical ICU. We used data from two periods: April 2015 to April 2016 (Phase one [P1]) and April 2017 to April 2018 (Phase [P2]). Between the two phases, six HFO devices were obtained and all ICU physicians were trained on the benefits and use of HFO, including three bibliographic sessions (literature review). The ICU nurses were also trained by the manufacturer in the use of the HFO device. No other changes in the respiratory management of patients (particularly no changes to ICU protocols) were implemented. A period of one year was chosen between the two phases because the implementation of practice changes is a long-term process and to allow for the inclusion of hospitalized patients from the same seasons. Patient records were accessed from April to July 2020 at the University Hospital of Angers. We first proceeded with the anonymization of the database before performing the data analysis (S1–S4 Tables).

## Participants

All adult patients (age ≥ 18 years) who were hospitalized in the intensive care unit, intubated, mechanically ventilated for a duration > 1 day, and extubated during one of the two study phases were included in our cohort.

## Variables

Patient characteristics, including age, body mass index, medical history (chronic obstructive pulmonary disease, arterial hypertension, sleep apnea syndrome, coronary artery disease), type of admission (medical, elective surgery, or emergency surgery), arterial partial pressure of oxygen ($PaO_2$)/fraction of inspired oxygen ($FiO_2$) ratio (in mmHg) and reason for mechanical ventilation on admission, shock on admission (if applicable; defined as the use of norepinephrine), and Simplified Acute Physiology Score (SAPS II) score were recorded [25]. The date of the first extubation as well as the $PaO_2$/$FiO_2$ ratio immediately before extubation were recorded. Additional information was recorded during the first 7 days following extubation, if applicable, including: use of noninvasive methods, time of implementation, physician indication ("prophylactic" to post-extubation ARF prophylaxis or "supportive" treatment of ARF to prevent reintubation to bridge until the underlying process), duration of use and settings (NIV: inspiratory pressure support, positive end expiratory pressure [PEEP], and $FiO_2$; HFO: flow rate and $FiO_2$) were recorded during the first 7 days following extubation.

The primary endpoint was the reintubation rate within the first 48 hours after extubation (excluding reintubation for surgery). Secondary endpoints included safety endpoints, such as the time to reintubation (time between extubation and reintubation), length of ICU stay, 28-day mortality, days without mechanical ventilation after 28 days (a value of 0 was assigned if the patient died) and the reintubation rate at day 7 post extubation.

## Statistical analysis

Quantitative data are expressed as medians (interquartile range) and were compared using the Mann–Whitney U test. Qualitative data are described as percentages and were compared using Fisher's exact test or chi-squared test if more than 1 degree of freedom.

For the primary endpoint, patients were separated into two groups, according to phase (P1 or P2). A multivariable analysis of the occurrence of reintubation within the 48 hours after extubation (excluding reintubation for surgery) was compared using Fisher's exact test. The

use of noninvasive methods described by frequency, indication, and combination of methods, were compared by Fisher's exact test. The survival analysis of the time to reintubation used a univariate log-rank test. A Kaplan Meier-type graphical representation was used to show the survival curves for each group. The other endpoints (time to reintubation, length of ICU stay, 28-day mortality, and days without mechanical ventilation during the first 28 days) were compared using the Mann–Whitney U test.

For some outcomes (i.e., reintubation rate at 48 hours, reintubation rate at 7 days, and survival analysis of the probability of reintubation censored at 7 days), multivariable analyses were performed. Reintubation rates were analyzed using a logistic regression, while the survival analysis of the probability of reintubation used a multivariable Cox cause-specific regression model. The association between the primary endpoint and risk factors for reintubation were expressed by a cause-specific hazard ratio (HR) considering the competitive risks of censored reintubation due to death, or patients discharged prior to the need for reintubation. For all models, we applied a selection process to choose among candidate predictor variables that had a univariable p-value <0.15.

A p-value <0.05 was considered statistically significant. Analyses were performed using R version 3.6.3 and SPSS Statistics version 24.0 (Armonk, NY: IBM Corp).

## Results

### Study participants

The patient selection flow chart is presented in Fig 1. Among the 302 patients included in the two cohorts, 290 were ultimately included in the analysis (181 in P1; 109 in P2). The majority were men, and median age of participants was 65 years ([50–74]; Table 1). The median SAPS was 48 ([37–59]; Table 1). The number of patients who were admitted following a recent surgery was 196 (67%). The clinical characteristics of the patients were similar between the two phases, except for a higher proportion of patients with coronary artery disease in P1, and a higher incidence of acute respiratory distress syndrome and shock on admission in P2 (Table 1).

The proportions of patients who received a noninvasive ventilation method were not significantly different between the two cohort phases (P1 [22.7%]; P2 [26.6%]; p = 0.480). The indication of noninvasive method was listed as prophylactic in 4.5% and 5.2% of patients in P1 and P2, respectively (p = 0.099). However, the noninvasive method was implemented significantly earlier after extubation in P2 (0 [0–7] versus 4 [0–23] hours; p = 0.009). HFO was used significantly more in P2 (83% of patients receiving a noninvasive method) versus 61% in P1 (p = 0.039; Table 1). NIV was preferentially prescribed as hour-long sessions, several times a day, whereas HFO was used almost continuously with an increase in daily HFO time initiated between the two phases (13.5 [7.0–18.7] versus 20.9 [17.7–22.5] hours per day per treated patient; p = 0.001; S1 Table). Based on these results, we confirmed a change for the use of noninvasive methods between the two periods with an earlier implementation of and extended post-extubation use of HFO in the second period (P2).

### Primary outcomes

In total, 24 patients (8.3%) were reintubated within the first 48 hours of extubation. The reintubation rate was significantly lower at 48 hours in the second phase of our cohort (20 [11.0%] versus 4 [3.7%] reintubations in P1 and P2, respectively; p = 0.028; Table 2).

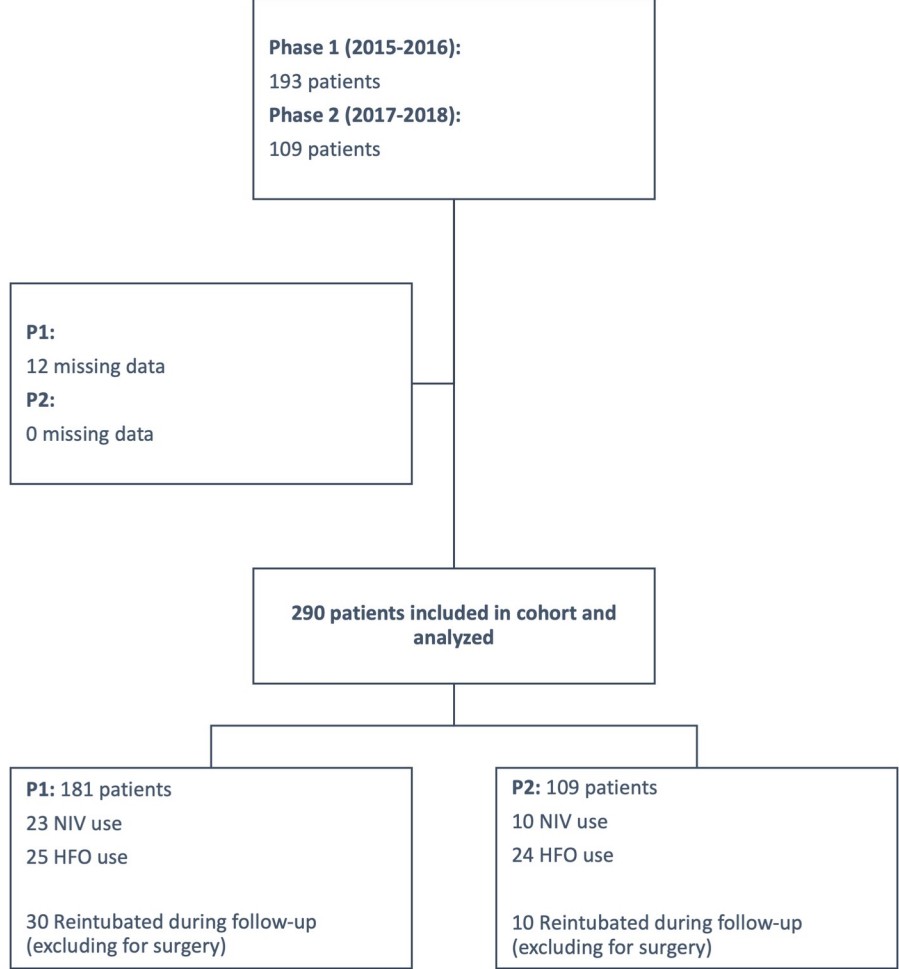

**Fig 1. Cohort selection flow chart.** NIV: Noninvasive ventilation; HFO: High-flow oxygen; P1: Phase 1; P2: Phase 2.

## Multivariable analysis

We conducted a multivariate analysis to explore the factors associated with successful extubation at 48 hours (Table 3). Considering the study variables that may influence successful extubation (patient comatose at admission, shock at admission, postoperative admission, and use of noninvasive ventilation method), Phase 2 was independently associated with less reintubations at 48 hours (OR 0.30 [95% CI: 0.08–0.92]; p = 0.034). This finding implies that changes in our usage of noninvasive methods between the two periods were not accompanied by a significant increase in the reintubation rate at 48 hours.

## Secondary outcomes

Forty patients (13.8%) were reintubated within the first 7 days after extubation. The main cause of reintubation in both phases was the occurrence of ARF (70% of reintubated patients in both phases). The median time to reintubation was not significantly earlier in P1 (1 [0–3] day versus 3 [1–5] for P1 and P2, respectively; p = 0.089). The reintubation rate at 7 days was not significantly different between the two phases (30 [16.6%] versus 10 [9.2%] for P1 and P2, respectively; p = 0.082; Table 2). We conducted a multivariate analysis to explore the factors

**Table 1. Baseline patient characteristics.**

| Characteristics | All (n = 290) | P1 (n = 181) | P2 (n = 109) | p-*value* |
|---|---|---|---|---|
| **Male sex** | 190 (65.5) | 121 (66.9) | 69 (63.3) | 0.610 |
| **Age, years** | 65 [50-74] | 65 [51-75] | 63 [47-72] | 0.218 |
| **BMI** | 26.8 [22.5–30.5] | 26.8 [22.3–29.9] | 26.8 [22.7–31.0] | 0.680 |
| **SAPS II** | 48 [37-59] | 48 [37-59] | 48 [39-59] | 0.836 |
| **Medical history** | | | | |
| COPD | 20 (6.9) | 10 (5.5) | 10 (9.2) | 0.242 |
| OSA | 19 (6.6) | 12 (6.6) | 7 (6.4) | 1.000 |
| Arterial Hypertension | 130 (44.8) | 84 (46.4) | 46 (42.2) | 0.543 |
| Coronary artery disease | 26 (9.0) | 22 (12.5) | 4 (3.7) | **0.018** |
| **Type of ICU admission** | | | | 0.908 |
| **Emergent surgery** | 171 (59.0) | 108 (59.7) | 63 (57.8) | |
| **Elective surgery** | 25 (8.6) | 16 (8.8) | 9 (8.3) | |
| **Medical:** | 94 (32.4) | 57 (31.5) | 37 (33.9) | |
| Non-operative trauma | 38 (13.1) | 19 (10.5) | 19 (17.4) | |
| Brain injury | 25 (8.6) | 19 (10.5) | 6 (5.5) | |
| Respiratory failure | 14 (4.8) | 9 (5.0) | 5 (4.6) | |
| Shock/Heart failure | 10 (3.4) | 9 (5.0) | 1 (0.9) | |
| Other | 7 (2.4) | 1 (0.6) | 6 (5.5) | |
| **Reason for ICU admission** | | | | |
| Pneumonia | 34 (11.7) | 19 (10.5) | 15 (13.8) | 0.453 |
| Acute pulmonary edema | 3 (1.0) | 3 (1.7) | 0 | 0.294 |
| ARDS | 13 (4.5) | 4 (2.2) | 9 (8.3) | **0.020** |
| Coma | 96 (33.1) | 60 (33.1) | 36 (33.0) | 1.000 |
| Shock | 95 (32.8) | 43 (23.8) | 52 (47.7) | $< \textbf{0.001}$ |
| **P/F at admission** | 288 [169-399] | 293 [172-418] | 277 [167-366] | 0.248 |
| **Extubation day** | 2 [1-8] | 3 [1-8] | 2 [1-8] | 0.488 |
| **P/F at extubation** | 287 [217-359] | 288 [216-366] | 283 [220-348] | 0.778 |
| **Use of a noninvasive method** | 70 (24.1) | 41 (22.7) | 29 (26.6) | 0.480 |
| **Time between extubation and noninvasive method implementation, hours** | 1 [0-12] | 4 [0-23] | 0 [0-7] | **0.009** |
| **NIV used** | 33 (11.4) | 23 (12.7) | 10 (9.2) | 0.146 |
| **Indication of NIV** | | | | 0.402 |
| **Prophylactic** | 10 (3.4) | 8 (4.4) | 2 (1.8) | |
| **Supportive** | 15 (5.2) | 9 (5.0) | 6 (5.5) | |
| **Other/not available** | 8 (2.7) | 6 (3.3) | 2 (1.8) | |
| **HFO used** | 49 (16.9) | 25 (13.8) | 24 (22.0) | **0.039** |
| **Indication of HFO** | | | | 0.085 |
| Prophylactic | 20 (6.9) | 7 (3.9) | 13 (11.9) | |
| Supportive | 29 (10.0) | 18 (9.9) | 11 (10.1) | |
| **Combination of NIV and HFO** | 12 (4.1) | 7 (3.9) | 5 (4.6) | 0.768 |

ARDS, acute respiratory distress syndrome; BMI, body mass index; COPD, chronic obstructive pulmonary disease; HFO, High-flow oxygenation; ICU, intensive care unit; NIV, noninvasive ventilation; OSA, obstructive sleep apnea; P1, Phase 1; P2, Phase 2; P/F, arterial partial pressure of oxygen (PaO2 in mmHg)/Fraction of inspired oxygen (FiO$_2$); SAPS, Simplified Acute Physiology Score.

n = percentage of group.

associated with reintubation within 7 post extubation days. The results showed that the study period was not independently associated with extubation failure during the first 7 days (OR 0.47 [CI: 0.18–1.09]; p = 0.093; S2 Table). Results observed at 48 hours were comparable to

**Table 2. Patient outcomes.**

| Outcomes | All (n = 290) | P1 (n = 181) | P2 (n = 109) | p-value |
|---|---|---|---|---|
| Reintubation within 48 hours[a], % | 24 (8.3) | 20 (11.0) | 4 (3.7) | **0.028** |
| Cause at 48 hours [a] | | | | 0.372 |
| Acute respiratory failure | 17 (5.9) | 13 (7.2) | 4 (3.7) | |
| Shock/cardiac arrest | 2 (0.7) | 2 (1.1) | 0 | |
| Neurologic failure | 5 (1.7) | 5 (2.8) | 0 | |
| Reintubation within 7 days [b] | 40 (13.8) | 30 (16.6) | 10 (9.2) | 0.082 |
| Cause at 7 days [b] | | | | 0.929 |
| Acute respiratory failure | 28 (9.7) | 21 (11.6) | 7 (6.4) | |
| Shock/cardiac arrest | 3 (1.0) | 2 (1.1) | 1 (0.9) | |
| Neurologic failure | 9 (3.1) | 7 (3.9) | 2 (1.8) | |
| Time to reintubation, days | 1 [0-3] | 1 [0-3] | 3 [1-5] | 0.089 |
| Days without ventilation on day 28 [c] | 24.0 [17.0–26.0] | 24.0 [17.0–26.0] | 24.0 [18.0–26.0] | 0.231 |
| ICU length of stay, days | 8.0 [4.0–17.0] | 8.0 [4.0–18.0] | 8.0 [4.0–17.0] | 0.766 |
| 28-days mortality, % | 16 (5.5) | 11 (6.1) | 5 (4.6) | 0.792 |

ICU, intensive care unit; P1, Phase 1; P2, Phase 2.

[a] Reintubation within 48 hours after extubation (excluding reintubation for surgery).

[b] Reintubation within 7 days after extubation (excluding reintubation for surgery).

[c] Value of 0 was affected if patient died during the 28 days after extubation.

n = percentage of group.

those at 7 days post extubation. There was not a significant increase in the reintubation rate in P2. Considering the factors that could influence successful extubation, this study period was not associated with reintubation.

The probability of reintubation censored at 7 days was not different between the two phases (adjusted hazard ratio of extubation failure in Phase 2 was 0.47 [95% CI: 0.22–1.04]; p = 0.061; Fig 2). The survival analyses calculated with an adjusted multivariable Cox model were not different between the two phases (S3 Table).

The use of noninvasive methods was independently associated with reintubation during the first 48 hours but not during the 7 days post extubation (p = 0.013 and p = 0.104, respectively; Table 3 and S2 Table). Reintubation was not specifically associated with either method when used alone (HFO alone versus NIV alone) at both 48 hours and at 7 days post extubation. However, after multivariable analysis and adjustment, the combined use of NIV and HFO in the same patients was independently associated with increased extubation failure during the first 48 hours (OR 15.18 [95% CI: 2.78–83.16]; p = 0.001; Table 3). This association between the combined use of NIV and HFO and increased extubation failure was also found during the first 7 days (OR 6.70 [95% CI: 1.46–31.43]; p = 0.013; S2 Table). The details of the combined use of NIV and HFO are presented in the S4 Table.

Finally, none of the other primary clinical outcomes (day-28 mortality, ICU length of stay, and ventilation-free days) were notably different between the two phases (Table 2).

## Discussion

In this single-center retrospective cohort study, we confirmed the safety of a practice change in the use of noninvasive methods, with the earlier and more extended post-extubation use of HFO in P2. This confirmation was not accompanied by a significant increase in the reintubation rate at neither 48 hours nor at 7 days, post extubation. Adjustments of factors that could influence successful extubation were conducted and the results revealed no association

**Table 3. Multivariate analysis of factors associated with reintubation within 48 hours, integrating study phase.**

| Characteristics | Reintubation at 48 hours[a] (n = 24) | Extubation Success (n = 266) | Univariate *P-value* | Multivariate Regression OR [95%CI][b] | p-value |
|---|---|---|---|---|---|
| **Male sex** | 16 (66.7) | 174 (65.4) | 1.000 | - | |
| **Age, years** | 64 [54-73] | 65 [50-74] | 0.949 | - | |
| **BMI** | 26.9 [24.4–33.2] | 26.7 [22.5–30.3] | 0.519 | - | |
| **SAPS II** | 45 [36-55] | 48 [38-59] | 0.218 | - | |
| **Medical history** | | | | | |
| **COPD** | 3 (12.5) | 17 (6.4) | 0.223 | - | |
| **OSA** | 1 (4.2) | 18 (6.8) | 1.000 | - | |
| **Arterial Hypertension** | 12 (50.0) | 118 (44.4) | 0.670 | - | |
| **Coronary artery disease** | 1 (4.2) | 25 (9.4) | 0.708 | - | |
| **Reason for ICU admission** | | | | | |
| **Pneumonia** | 1 (4.2) | 33 (12.4) | 0.331 | - | |
| **Acute pulmonary edema** | 0 | 3 (1.1) | 1.000 | - | |
| **ARDS** | 0 | 13 (4.9) | 0.610 | - | |
| **Coma** | 12 (50.0) | 84 (31.6) | 0.074 | 2.36 [0.88–6.56] | 0.086 |
| **Shock** | 3 (12.5) | 92 (34.6) | **0.039** | 0.47 [0.10–1.61] | 0.240 |
| **Postoperative admission** | 21 (52.5) | 170 (70.0) | **0.044** | 0.76 [0.30–2.00] | 0.577 |
| **Postoperative day** | 0 [0-0] | 0 [0-0] | 0.518 | - | |
| **P/F at admission** | 269 [184–377] | 293 [169–401] | 0.571 | - | |
| **Extubation day** | 3 [1-17] | 2 [1-8] | 0.262 | - | |
| **P/F at extubation** | 275 [206-337] | 288 [218-366] | 0.346 | - | |
| **Phase 2** | 4 (16.7) | 105 (39.5) | **0.028** | **0.30 [0.08–0.92]** | **0.034** |
| **Prophylactic strategy** | 4 (16.7) | 24 (9.0) | 0.268 | | |
| **Noninvasive method** | | | **<0.001** | | **0.013** |
| **None** | 15 (62.5) | 205 (77.1) | Reference | - | - |
| **NIV alone** | 2 (8.3) | 19 (7.4) | 0.065 | 1.46 [0.20–6.73] | 0.663 |
| **HFO alone** | 2 (8.3) | 35 (13.2) | 0.750 | 0.90 [0.13–3.85] | 0.897 |
| **Combination** | 5 (20.8) | 7 (2.6) | **<0.001** | 15.18 [2.78–83.16] | **0.001** |

ARDS, acute respiratory distress syndrome; BMI, body mass index; COPD, chronic obstructive pulmonary disease; HFO, high-flow oxygen; ICU, intensive care unit; NIV, noninvasive ventilation; OSA, obstructive sleep apnea; P/F, arterial partial pressure of oxygen ($PaO_2$)/Fraction of inspired oxygen ($FiO_2$); SAPS, Simplified Acute Physiology Score.

[a] Reintubation during 48 post-extubation hours (excluding reintubation for surgery).

[b] Odds ratio [95% Confidence Interval].

n = percentage of phase.

between extubation failure and the second phase of the study. Use of noninvasive methods has traditionally raised safety concerns [23], mainly due to the potentially increased risk of poorer outcomes resulting from delayed reintubation. Our results do not seem to support these concerns. The median time to reintubation tended to be earlier in P1 (1 [0–3] versus 3 [1–5] days in P1 and P2, respectively; p = 0.089), with no difference in 28-day mortality (p = 0.792), ICU length of stay (p = 0.766), or days without ventilation (p = 0.231).

Studies comparing data on NIV and HFO are available; however, they are inconclusive about oxygenation, work of breathing, and need for intubation [26]. In a randomized controlled trial, Hernandez et al. [20] showed that in a mix of high-risk medical and surgical, post-extubation patients, HFO was similar to NIV in the prevention of reintubation and post-extubation respiratory failure. In another randomized controlled trial with cardiothoracic surgical

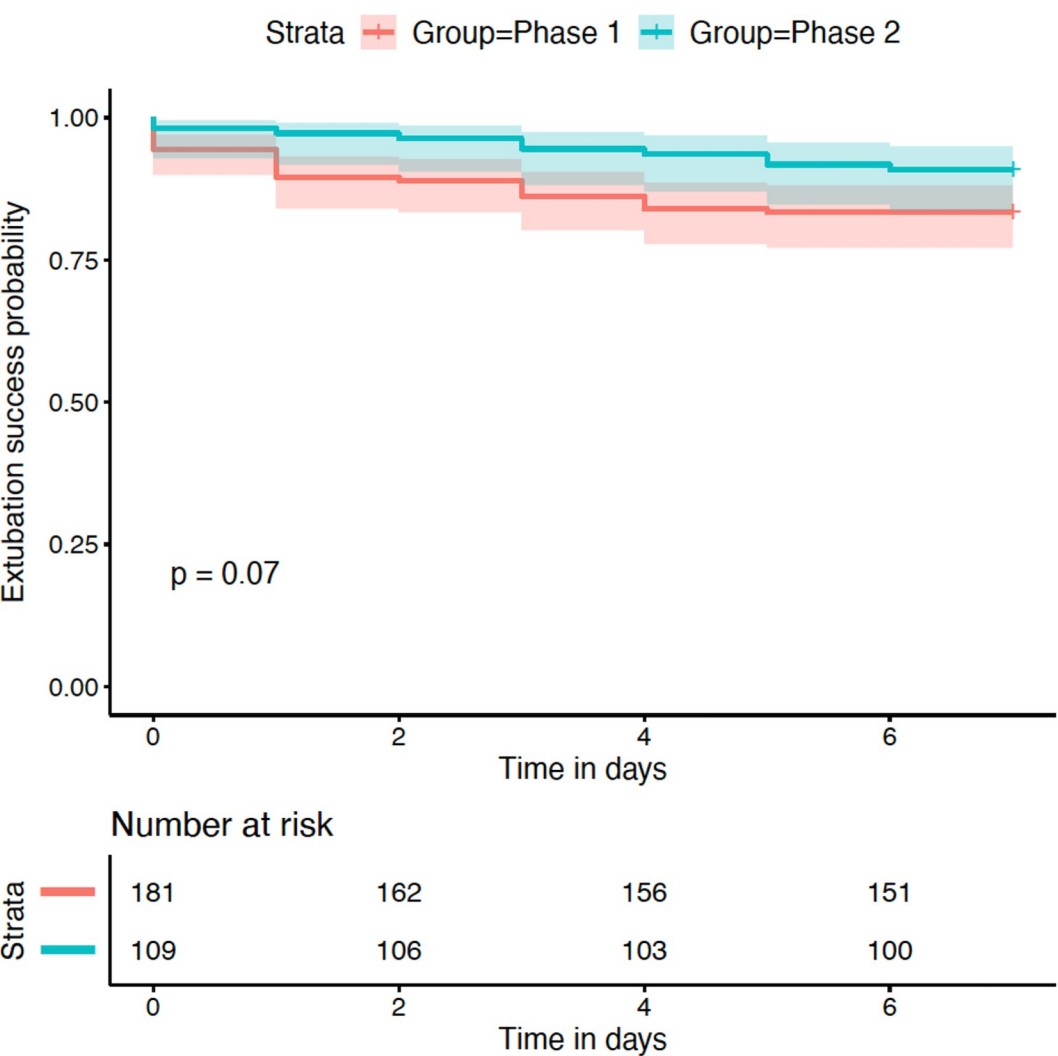

**Fig 2. Kaplan–Meier curves showing extubation success within 7 days according to study phase.** P-value obtained with the log-rank test.

patients, with or at risk for ARF, Stéphan et al. [21] showed that the use of continuous HFO did not result in higher rates of treatment failure than did intermittent NIV (reintubation in 14.0% versus 13.7%; p = 0.99). Results of previous published studies were pooled in a meta-analysis comparing the effects of HFO and NIV after extubation, and there was no significant difference in the rates of treatment failure (OR 0.96 [95% CI: 0.75–1.24]; p = 0.77) and reintubation (OR 1.00 [95% CI: 0.76–1.32]; p = 0.98) [21].

We chose to assess the reintubation rate at 48 hours as a safety endpoint because extubation failure is currently defined as the need to reintubate within 48 hours after extubation [8,9]. In our study, the overall post-extubation rate in ARF within 48 hours was approximately 8%, which is similar to the overall rate reported at 48–72 hours in previous randomized trials assessing noninvasive methods [3,19]. The rate of reintubation due to non-respiratory causes was greater than 30% in our study. However, this finding is probably related to the high proportion of brain injury patients in our ICU population (33.1% of comatose patients at admission). In these patients, reintubation for neurological failure is obviously more frequent [7]. Furthermore, in the context of neurological failure, the use of a noninvasive method is unlikely

to prevent reintubation (22.5% of reintubations were for neurologic failure). Patient selection with less risk factors for neurologic failure should be considered to potentially maximize the benefit of noninvasive methods to prevent reintubation.

In our study, we observed a decrease of 70% in the reintubation rate at 48 hours after extubation, but not at 7 days. Indeed, the reintubation rate we observed at 7 days (13.8% overall) was similar to those reported in previous studies [27]. Recently, Thille et al. proposed to extend the definition of extubation failure to 7 days after extubation when using noninvasive methods [28]. Our results suggest that an evaluation of extubation failure at 7 days is more clinically appropriate than at 48 hours.

In our cohort, HFO therapy lasted longer than NIV (daily therapy duration and number of days). This may be related to a poorer tolerance of NIV than HFO and improved comfort under HFO [3]. Patient care can be improved through increased respiratory comfort; however, these factors were not extensively evaluated in our study, warranting the need for future research.

The combination of NIV and HFO has been proposed to prevent post-extubation ARF. Thille et al. [27] showed that the combination of NIV and HFO after extubation significantly decreased the risk of reintubation compared to HFO alone. We observed that the reintubation rate independently increased in the subgroup of patients treated with a combination of NIV and HFO (OR 15.18 (95% CI:2.78–83.16]; p = 0.001; Table 3). The proportion of surgical patients was higher in our cohort. The differences in the pathophysiology of ARF between surgical and medical patients could be partially explained by this discrepancy. In our cohort, the combined use of NIV and HFO was low (4.1%) in both phases. We can assume that the combination of therapy was used on the more serious patients, explaining the increase in extubation failure in this sub-group. The small number of patients does not allow for definitive conclusions about the combined use of NIV and HFO. Further studies focusing on the safety of using a combination noninvasive methods to prevent or treat ARF are needed.

To our knowledge, this is the first study to describe the safety and evolution of the use of noninvasive methods in a real-life setting. Many studies that have focused on the post-operative use of noninvasive methods were limited to the comparison of NIV and HFO; however, few have included patients who did not receive any of these methods. Lastly, in our study, physicians were free to choose the strategy they considered most appropriate. The changes in the use of non-invasive methods are therefore likely to be spontaneously observed in any ICU.

This study has several limitations. First, given the retrospective and observational design of this study, we cannot establish any causal relationship. Second, missing data about the use of noninvasive ventilation or reintubation status might have influenced the results. Third, the lack of statistical power did not allow for conclusions about our second endpoint criteria (reintubation at 7 days). Lastly, the single-center design may have exacerbated the selection bias.

## Conclusions

In this single-center retrospective cohort study, earlier implementation of noninvasive ventilation methods and increased use of HFO were observed between the two phases. Reintubation rates did not increase in the first 48 hours or at 7 days post extubation with the implementation of practice changes in the post-extubation use of NIV and HFO in our surgical ICU. These results suggest that these practice changes are safe and effective.

## Supporting information

**S1 Table. Characteristics of use and setting of noninvasive methods.** FiO$_2$: Fraction inspired of oxygen. NIV: Noninvasive ventilation. HFO: High-flow oxygenation.
(DOCX)

**S2 Table. Multivariate analysis of factors associated with reintubation within 7 days, integrating study phase.** n (% of group). Odd ratio [95% Confidence Interval]: OR[95%IC]. BMI: Body mass index. COPD: Chronic obstructive pulmonary disease. OSA: Obstructive sleep apnea. ICU: Intensive care unit. ARDS: Acute respiratory distress syndrome. P/F: Arterial partial pressure of oxygen (PaO2 in mmHg)/Fraction inspired of oxygen (FiO$_2$). [a] Reintubation within 7 days after extubation (excluding reintubation for surgery).
(DOCX)

**S3 Table. Multivariable Cox analysis of factors associated with reintubation integrating study phase.** n (% of group). Hazard ratio [95% Confidence Interval]: HR[95%IC]. -: Not available. BMI: Body mass index. COPD: Chronic obstructive pulmonary disease. OSA: Obstructive sleep apnea. ICU: Intensive care unit. P/F: Arterial partial pressure of oxygen (PaO$_2$)/Fraction inspired of oxygen (FiO$_2$). [a] Reintubation within 7 days after extubation (excluding reintubation for surgery).
(DOCX)

**S4 Table. Detail of combined use of NIV and HFO.** No statistical comparisons were made due to the small number of patients.
(DOCX)

**S1 Data.**
(XLSX)

## Author Contributions

**Conceptualization:** Stanislas Abrard, Sigismond Lasocki.

**Data curation:** Stanislas Abrard, Maxime Léger.

**Formal analysis:** Stanislas Abrard, Maxime Léger.

**Investigation:** Stanislas Abrard, Lorine Jean.

**Methodology:** Stanislas Abrard, Maxime Léger, Sigismond Lasocki.

**Project administration:** Sigismond Lasocki.

**Resources:** Pauline Dupré.

**Supervision:** Sigismond Lasocki.

**Validation:** Emmanuel Rineau, Maxime Léger, Sigismond Lasocki.

**Visualization:** Stanislas Abrard, Lorine Jean.

**Writing – original draft:** Stanislas Abrard, Lorine Jean.

**Writing – review & editing:** Emmanuel Rineau, Maxime Léger, Sigismond Lasocki.

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
