## [Decision Letter · Decision Letter 0]

19 Oct 2020

PONE-D-20-29871

Effect of changes in the use of noninvasive ventilation and high flow oxygen therapy on reintubation in a surgical intensive care unit. A Retrospective Cohort Study

PLOS ONE

Dear Dr. Abrard,

Thank you for submitting your manuscript to PLOS ONE. After careful consideration, we feel that it has merit but does not fully meet PLOS ONE’s publication criteria as it currently stands. Therefore, we invite you to submit a revised version of the manuscript that addresses the points raised during the review process.

The authors should note that one of the reviewers felt there was extensive need for editing, specifically asking for more details on study design and results.  I would also add that is unclear what processes occurred between phase 1 and phase 2 that resulted in more HFO2 use: education amongst physicians? education of respiratory therapists?  New protocols/policies?  These need to be spelled out.  Finally, as the reviewer pointed out there has been randomized studies showing no difference in outcomes between NIV and HFO2; the authors need to discuss why there results are different and whether clinically relevant given randomized trial data. 

We look forward to receiving your revised manuscript.

Kind regards,

James Andrew Rowley

Academic Editor

PLOS ONE

Journal Requirements:

2. In the ethics statement in the manuscript and in the online submission form, please provide additional information about the patient records used in your retrospective study, including: a) whether all data were fully anonymized before you accessed them and/or whether the IRB or ethics committee waived the requirement for informed consent;; b) the date range (month and year) during which patients' medical records were accessed; c) the source of the medical records analyzed in this work (e.g. hospital, institution or medical center name). If patients provided informed written consent to have data from their medical records used in research, please include this information.

Reviewers' comments:

Reviewer's Responses to Questions

**Comments to the Author**

1. Is the manuscript technically sound, and do the data support the conclusions?

Reviewer #1: Partly

Reviewer #2: Yes

2. Has the statistical analysis been performed appropriately and rigorously? 

Reviewer #1: I Don't Know

Reviewer #2: Yes

3. Have the authors made all data underlying the findings in their manuscript fully available?

Reviewer #1: Yes

Reviewer #2: Yes

4. Is the manuscript presented in an intelligible fashion and written in standard English?

Reviewer #1: Yes

Reviewer #2: Yes

5. Review Comments to the Author

Reviewer #1: This paper needs to be heavily edited and revised, including adding more explanation for the difference observed between phase 1 and phase 2 and between the patients and practice improvements in phase 1 and phase 2, which would require more detail in the study design, results, and discussion. Although this study subject was very interesting and clearly involved a lot of work with data collection and analysis, this paper wasn't strong enough to change current practice, especially since there have been 2 multicenter publications showing no difference in HFO and NIV for reintubation.

Reviewer #2: Data in the abstract and data in the tables don't align or just isn't clear. For example, in the abstract the HFO use in P2 is 24 with 83% and in Table 1 P2 has 24 with percent of 22%. In looking at the table it looks like the 2nd time period had less HFO use (greater percentage is true) and then had a decreased reintubation rate at 48 hour. They stated that there was a increase in education between the 2 time groups about HFO and while statistically significant the clinical significant of 1 patient is questionable.

In addition about 30% of the patient in the study are not postoperative patients. If want to make an assessment about HFO in SICU patient population it should all be surgical or at least define where these nonoperative patients come from (i.e. are they non-operative trauma, head injury, etc.)

There are several questionable typing errors of the p being reported at p<.15 on line 143, which I'm assuming should be p<0.05?

6. PLOS authors have the option to publish the peer review history of their article (what does this mean?). If published, this will include your full peer review and any attached files.

Reviewer #1: No

Reviewer #2: No

---

## [Author Response · Author response to Decision Letter 0]

2 Dec 2020

Dear Editor,

I am pleased to submit this revised version of our manuscript (Ref: PONE-S-20-37260) entitled “Effect of changes in the use of noninvasive ventilation and high flow oxygen therapy on reintubation in a surgical intensive care unit. A Retrospective Cohort Study.” for possible publication in Plos One.

You will find below the point-by-point response to reviewers' comments. The changes are reported in red in the revised text.

We, authors of PONE-S-20-37260, thank you for your time and consideration. Reviewers’ comments helped us revise the manuscript, and we believe that content and clarity are now improved in this revised version. 

Sincerely yours,

Stanislas Abrard

---

## [Decision Letter · Decision Letter 1]

22 Dec 2020

PONE-D-20-29871R1

Effect of changes in the use of noninvasive ventilation and high flow oxygen therapy on reintubation in a surgical intensive care unit. A Retrospective Cohort Study

PLOS ONE

Dear Dr. Abrard,

Thank you for submitting your manuscript to PLOS ONE. After careful consideration, we feel that it has merit but does not fully meet PLOS ONE’s publication criteria as it currently stands. Therefore, we invite you to submit a revised version of the manuscript that addresses the points raised during the review process.

Please note two things:

1. I have reviewed Reviewer #1s detailed comments and agree that more work needs to be done for clarity. 

2. Reviewer #2 reviewed your revision outside of Editorial Manager and felt that you have responded to her comments in an appropriate manner.

Therefore, key that you respond carefully to Reviewer #1.

We look forward to receiving your revised manuscript.

Kind regards,

James Andrew Rowley

Academic Editor

PLOS ONE

Reviewers' comments:

Reviewer's Responses to Questions

**Comments to the Author**

1. If the authors have adequately addressed your comments raised in a previous round of review and you feel that this manuscript is now acceptable for publication, you may indicate that here to bypass the “Comments to the Author” section, enter your conflict of interest statement in the “Confidential to Editor” section, and submit your "Accept" recommendation.

Reviewer #1: All comments have been addressed

2. Is the manuscript technically sound, and do the data support the conclusions?

Reviewer #1: Partly

3. Has the statistical analysis been performed appropriately and rigorously? 

Reviewer #1: Yes

4. Have the authors made all data underlying the findings in their manuscript fully available?

Reviewer #1: Yes

5. Is the manuscript presented in an intelligible fashion and written in standard English?

Reviewer #1: No

6. Review Comments to the Author

Reviewer #1: see attachment for the rest of the comments. 1. There needs further revision for clarity of thought and free of grammatical errors. for example, in line 59, "pathophysiology" is not used correctly here. It is a term to describe the physiologic mechanism behind symptoms so should not be used to describe a list of clinical manifestations or symptoms. there should also be a "and" before cardiac dysfunction in line 60. In line 67, the use of the word "curative" is also odd because NIV doesn't "cure" tracheal reintubation, since it is a supportive measure to bridge until the underlying process is addressed so should be referred to as "preventative". In line 69, the phrase "a more recent method" is unclear and refers to that is a more recent developed technology compared to NIV but I think only a reader who would know that context would understand this, so instead of "recent method" should be changed to "a more recently developed technology". But that sentence doesn't clearly distinguish the difference between NIV and HFO. 2. Results section needs major revising and recommendations were given. 3. Discussion section needs major revising, especially in the context of how your results fit into the current literature especially the two meta-analyses. 4. Conclusion: needs major revising--the wording in the sentence is not only confusing but also misleading. It suggests that using earlier use of non-invasive and HFO will decrease re-intubation, but your study is not a prospective study on the timing of NIV and HFO on reintubation rates.

7. PLOS authors have the option to publish the peer review history of their article (what does this mean?). If published, this will include your full peer review and any attached files.

Reviewer #1: No

---

## [Author Response · Author response to Decision Letter 1]

1 Feb 2021

Reviewer #1 comments:

1. There needs further revision for clarity of thought and free of grammatical errors. for example, in line 59, "pathophysiology" is not used correctly here. It is a term to describe the physiologic mechanism behind symptoms so should not be used to describe a list of clinical manifestations or symptoms. there should also be a "and" before cardiac dysfunction in line 60.

An Editing Service edited this manuscript. We replaced « pathophysiology » in line 59 by « causes ». 

In line 67, the use of the word "curative" is also odd because NIV doesn't "cure" tracheal reintubation, since it is a supportive measure to bridge until the underlying process is addressed so should be referred to as "preventative".

Thank you for allowing us to clarify this point. Of course, the NIV does not treat tracheal intubation. We wanted to differentiate between two situations well described by the guidelines (Official ERS/ATS clinical practice guidelines: noninvasive ventilation for acute respiratory failure. Bram Rochwerg et al. Eur Resp J 2017): NIV for treatment of ARF in prevention of reintubation (patient with ARF criteria) named “curative” NIV in our previous manuscript and NIV for prevent post-extubation ARF (patient without AFR criteria) named “preventive” NIV in our previous manuscript. To improve clarity of our manuscript, in the revised version we named “prophylactic” NIV for prevent post-extubation ARF (patient without AFR criteria) and “supportive treatment by NIV” NIV for treatment of ARF in prevention of reintubation (patient with ARF criteria) to bridge until the underlying process.

In line 69, the phrase "a more recent method" is unclear and refers to that is a more recent developed technology compared to NIV but I think only a reader who would know that context would understand this, so instead of "recent method" should be changed to "a more recently developed technology". But that sentence doesn't clearly distinguish the difference between NIV and HFO. 

This term has been changed and precisions have been added.

In lines 83 & 84, connect the reason to your unit's change in practice more clearly to the benefits as described in lines 73. Also, it would be more clear to define specific worsening outcomes in line 79, because patient safety should then be addressed as part of your primary objectives and results. In light of the two meta-analyses and studies of prospective randomized controlled studies, at the most your retrospective study can show no worsening of patients but would not be able to prove HFO is superior. 

The benefits described in the introduction have been more clearly linked to the reason of our unit practice changes.

The introduction and discussion paragraphs have been greatly revised regarding patient safety during this change in practice and no worsening.

In line 109, this sentence "we proceeded to anonymization of database before data analysis" is awkwardly phrased and needs re-writing. 

This sentence has been reformulated.

Line 113 the term "extubated alive" is a strange phrase. Would recommend removing "alive". Does this mean patients who were deceased post-re-intubation were excluded? If so, this would potentially skew your results to showing falsely favorable outcomes because the poor outcomes were already removed from analysis, therefore needs to be stated as a major limitation of the study. 

We have removing the term “alive”. No, only patients deceased before an first extubation were excluded (never extubated).

The line 122-126 is also awkwardly phrased and need re-writing.

These sentences have been reformulated.

Line 128-131 needs re-writing for clarity. e.g. secondary endpoints included the reintubation rate on day 7 post-extubation,....

These sentences have been reformulated.

In line 220, “in respectively” is a phrase that doen’t make sense grammatically. 

This sentence has been reformulated.

Line 222, “at 7th day” also is a strange phrase-better to say “with maintaining successful extubation by post-extubation day 7.”

This sentence has been reformulated.

Line 224-226 was confusingly and the point was lost. Please re-phrase. 

These sentences have been reformulated.

Line 237 to 239, it is not enough to say “there is an association” between this and extubation failure but you need to explicitly say and interpret for the reader what the association is actually meaning (you state this in your discussion section but need to write something in the results too), with context of your variables. 

These sentences have been reformulated.

2. In the discussion section, you need to refer to the primary objective which is determining the re-intubation rates with the change in practice but also if the change in practice caused a null, positive, or negative impact on patient care. A lot of the first paragraph are sentences that should be in the result section. The discussion should start by discussing how your main results are meeting or not meeting your objective and the societal or patient impact of your results, such as how you go into this in your second paragraph.

The conclusion has been entirely rewrite following your advices.

Line 271, you wrote “specifically associated specifically”. This sentence needs to be re-written. 

This sentence has been rewrite.

Line 273, “the” needs to be inserted before combination.

Modified

Discussion section needs major revising, especially in the context of how your results fit into the current literature especially the two meta-analyses.

Please go through the discussion more thoroughly with significant revision and editing.

The discussion paragraph has been greatly revised integrating the two meta-analyses and edited by an Editing Service.

3. Conclusion: needs re-writing for clarity and grammar. This sentence suggests a though as derived from the paper, but has to be able to stand alone so that a reader would understand the main point of the paper prior to reading the details in the paper. I think the main point is actually there were no worsened re-intubation rates with the change of practice of increasing the use of HFO and NIV post-extubation in a surgical icu. The wording in the sentence is not only confusing but also misleading. It suggests that using earlier use of non-invasive and HFO will decrease re-intubation, but your study is not a prospective study on the timing of NIV and HFO on reintubation rates.

The conclusion has been entirely rewrite following your advices.

---

## [Decision Letter · Decision Letter 2]

15 Feb 2021

PONE-D-20-29871R2

Safety of changes in the use of noninvasive ventilation and high flow oxygen therapy on reintubation in a surgical intensive care unit:A retrospective cohort study

PLOS ONE

Dear Dr. Abrard,

Thank you for submitting your manuscript to PLOS ONE. After careful consideration, we feel that it has merit but does not fully meet PLOS ONE’s publication criteria as it currently stands. Therefore, we invite you to submit a revised version of the manuscript that addresses the points raised during the review process.

The reviewer still raised some issues that required to be addressed.

We look forward to receiving your revised manuscript.

Kind regards,

Yu Ru Kou, PhD

Academic Editor

PLOS ONE

Reviewers' comments:

Reviewer's Responses to Questions

**Comments to the Author**

1. If the authors have adequately addressed your comments raised in a previous round of review and you feel that this manuscript is now acceptable for publication, you may indicate that here to bypass the “Comments to the Author” section, enter your conflict of interest statement in the “Confidential to Editor” section, and submit your "Accept" recommendation.

Reviewer #1: All comments have been addressed

2. Is the manuscript technically sound, and do the data support the conclusions?

Reviewer #1: Yes

3. Has the statistical analysis been performed appropriately and rigorously? 

Reviewer #1: Yes

4. Have the authors made all data underlying the findings in their manuscript fully available?

Reviewer #1: Yes

5. Is the manuscript presented in an intelligible fashion and written in standard English?

Reviewer #1: Yes

6. Review Comments to the Author

Reviewer #1: This draft has significantly improved. There are still a few areas of ambiguous language that require clarification:

1.Line 248 to 252: The authors stated there was an association between the combination of NIV and HFO with extubation failure. For the ease of the reader, it would be better to qualify this association such as adding “increased” before “extubation failure” in line 249 and again adding "increased" before extubation in line 251.

2. Line 292-294: Recommend to rephrase this because the meaning was unclear. I am guessing on authors' intention, but one suggestion could be: “The underlying causes of neurologic failure or shock vary so not all of the contributing factors were included, such as over-sedation causing hypercapnia or hypovolemia causing shock. Patient selection with less risk factors for neurologic failure or shock should be considered to potentially maximize the benefit of noninvasive methods to prevent reintubation.”

3. Line 313: The phrase “without evolution over the phases” was confusing. Can the authors instead of this phrase just state more simply “in both phases.”

4. Line 314: The phrase “proposed to the more serious patients” may not be quite accurate, because proposed means to suggest but did not actually employ this method. Instead, are the authors meaning “used on the more serious patients.”

7. PLOS authors have the option to publish the peer review history of their article (what does this mean?). If published, this will include your full peer review and any attached files.

Reviewer #1: **Yes: **Sarah J. Lee

---

## [Author Response · Author response to Decision Letter 2]

17 Feb 2021

Dear Editor,

I am pleased to submit this revised version of our manuscript (Ref: PONE-S-20-29871R3) entitled “Safety of changes in the use of noninvasive ventilation and high flow oxygen therapy on reintubation in a surgical intensive care unit: A retrospective cohort study.” for possible publication in Plos One.

You will find below the point-by-point response to reviewers' comments. The changes are reported in red in the revised text.

Sincerely yours,

Stanislas Abrard

Q 1.Line 248 to 252: The authors stated there was an association between the combination of NIV and HFO with extubation failure. For the ease of the reader, it would be better to qualify this association such as adding “increased” before “extubation failure” in line 249 and again adding "increased" before extubation in line 251.

A: Thank to this comment. We have adding "increased" before "extubation" in lines 249 and 251.

Q 2. Line 292-294: Recommend to rephrase this because the meaning was unclear. I am guessing on authors' intention, but one suggestion could be: “The underlying causes of neurologic failure or shock vary so not all of the contributing factors were included, such as over-sedation causing hypercapnia or hypovolemia causing shock. Patient selection with less risk factors for neurologic failure or shock should be considered to potentially maximize the benefit of noninvasive methods to prevent reintubation.”

A: Thank you for your suggestion.

Q 3. Line 313: The phrase “without evolution over the phases” was confusing. Can the authors instead of this phrase just state more simply “in both phases.”

A: We have made the change.

Q 4. Line 314: The phrase “proposed to the more serious patients” may not be quite accurate, because proposed means to suggest but did not actually employ this method. Instead, are the authors meaning “used on the more serious patients.”

A: We have made the change.

---

## [Decision Letter · Decision Letter 3]

2 Mar 2021

PONE-D-20-29871R3

Safety of changes in the use of noninvasive ventilation and high flow oxygen therapy on reintubation in a surgical intensive care unit: A retrospective cohort study

PLOS ONE

Dear Dr. Abrard,

Thank you for submitting your manuscript to PLOS ONE. After careful consideration, we feel that it has merit but does not fully meet PLOS ONE’s publication criteria as it currently stands. Therefore, we invite you to submit a revised version of the manuscript that addresses the points raised during the review process.

The reviewer had one minor editorial suggestion.

We look forward to receiving your revised manuscript.

Kind regards,

Yu Ru Kou, PhD

Academic Editor

PLOS ONE

Journal Requirements:

Reviewers' comments:

Reviewer's Responses to Questions

**Comments to the Author**

1. If the authors have adequately addressed your comments raised in a previous round of review and you feel that this manuscript is now acceptable for publication, you may indicate that here to bypass the “Comments to the Author” section, enter your conflict of interest statement in the “Confidential to Editor” section, and submit your "Accept" recommendation.

Reviewer #1: All comments have been addressed

2. Is the manuscript technically sound, and do the data support the conclusions?

Reviewer #1: Yes

3. Has the statistical analysis been performed appropriately and rigorously? 

Reviewer #1: Yes

4. Have the authors made all data underlying the findings in their manuscript fully available?

Reviewer #1: Yes

5. Is the manuscript presented in an intelligible fashion and written in standard English?

Reviewer #1: Yes

6. Review Comments to the Author

Reviewer #1: Recommend removing the sentence in lines 292-294 "The underlying causes of neurologic failure or shock vary so not all of the contributing factors were included, such as over-sedation causing hypercapnia or hypovolemia causing shock." This sentence's grammar and meaning are still unclear.

7. PLOS authors have the option to publish the peer review history of their article (what does this mean?). If published, this will include your full peer review and any attached files.

Reviewer #1: No

---

## [Author Response · Author response to Decision Letter 3]

8 Mar 2021

Reviewer #1: Recommend removing the sentence in lines 292-294 "The underlying causes of neurologic failure or shock vary so not all of the contributing factors were included, such as over-sedation causing hypercapnia or hypovolemia causing shock." This sentence's grammar and meaning are still unclear.

Answer: This sentence has been deleted.

---

## [Editor Report · Decision Letter 4]

10 Mar 2021

Safety of changes in the use of noninvasive ventilation and high flow oxygen therapy on reintubation in a surgical intensive care unit: A retrospective cohort study

PONE-D-20-29871R4

Dear Dr. Abrard,

We’re pleased to inform you that your manuscript has been judged scientifically suitable for publication and will be formally accepted for publication once it meets all outstanding technical requirements.

Kind regards,

Yu Ru Kou, PhD

Academic Editor

PLOS ONE
---

## [Editor Report · Acceptance letter]

12 Mar 2021

PONE-D-20-29871R4 

Safety of changes in the use of noninvasive ventilation and high flow oxygen therapy on reintubation in a surgical intensive care unit: A retrospective cohort study 

Dear Dr. Abrard:

I'm pleased to inform you that your manuscript has been deemed suitable for publication in PLOS ONE. Congratulations! Your manuscript is now with our production department. 

Kind regards, 

on behalf of

Dr. Yu Ru Kou 

Academic Editor

PLOS ONE